# Microbiological quality assessment of five common foods sold at different points of sale in Burkina-Faso

**Muller Kiswendsida Abdou Compaore**[1,2]\*, **Stéphane Dissinviel Kpoda**[1,3☯], **Raoul Bazoin Sylvain Bazie**[1,2☯], **Marcelline Ouedraogo**[1‡], **Mahamady Valian**[1‡], **Marie-Louise Gampene**[1‡], **Alphonse Yakoro**[1‡], **Fulbert Nikiema**[1‡], **Asseto Belemlougri**[1‡], **Naamwin-So-Bawfu Romaric Meda**[1☯], **Naa-Imwine Stanislas Dimitri Meda**[1‡], **Souleymane Sanon**[4☯], **Moumouni Bande**[1,5☯], **Hervé Hien**[4☯], **Nicolas Barro**[2☯], **Elie Kabre**[1,5☯]

1 Laboratoire National de Santé Publique (LNSP), Ouagadougou, Burkina Faso, 2 Université Joseph KI-ZERBO, Laboratoire de Biologie moléculaire, d'Epidémiologie et de Surveillance des agents Transmissibles par les Aliments (LaBESTA), Centre de Recherche en Sciences Biologiques Alimentaires et Nutritionnelles (CRSBAN), École Doctorale Sciences et Technologies, Ouagadougou, Burkina Faso, 3 Université Joseph KI-ZERBO, Centre Universitaire de Ziniaré, Ouagadougou, Burkina Faso, 4 Institut National de Santé Publique (INSP), Ouagadougou, Burkina Faso, 5 Université Joseph KI-ZERBO, Unité de Formation et de la Recherche en Sciences de la Santé, Ouagadougou, Burkina Faso

☯ These authors contributed equally to this work.
‡ MO, MV, MLG, AY, FN, AB, and NSDM also contributed equally to this work.
\* mullercompaore@yahoo.fr

**Data Availability Statement:** All relevant data are within the paper and its Supporting information files.

## Abstract

The aim of the present study was to assess the microbial quality of five ready-to-eat food such as bread, pasta, rice with sauce, beans and milk sold in five localities of Burkina Faso namely, Ouagadougou, Bobo-Dioulasso, Dakola, Cinkansé and Niangoloko. One hundred and one samples were collected and microbial quality were assessed by evaluating the food hygiene indicators such as total aerobic mesophilic flora, total coliforms, thermotolerant coliforms, yeast and mould. Food safety indicators such as *Escherichia coli*, *Salmonella*, coagulase-positive staphylococci, *Clostridium perfringens* and *Bacillus cereus* were also tested for contamination. Samples were tested according to ISO guidelines for all parameters.

The results showed that 74 (73.27%) of samples were satisfactory while 15 (14.85%) were acceptable and 12 (11.88%) were not satisfactory according to international standards. Among the food safety indicators sought, *Escherichia coli* was detected in two samples and *Bacillus cereus* in four samples. Most of the analyzed food exhibited good hygiene behavior within the acceptable limits and the highest of not satisfactory rate was observed in milk powder and rice with sauce. Ouagadougou samples recorded the highest number of not satisfactory samples.

Despite the general quality was satisfactory, the presence of specific microorganisms such as coliforms is indicative of the poor hygiene surrounded these foods. It is therefore necessary to train and follow up the vendors in the handling of equipment, hand-washing practices and selling environment hygiene for better improvement of the quality of the street foods.

**Funding:** This research project is jointly implemented by the National Institute of Public Health and the National Public Health Laboratory. It was fully funded by the European Union (EU) under number FED/2019/407-596 The funders had no role in study design, data collection and analysis, decision to publish, or preparation of the manuscript. Grant Recipient: Pr. Elie KABRE.

**Competing interests:** The authors have declared that no competing interests exist.

## Introduction

Food quality is always a concern when intended to human consumption. Food-borne diseases have been increasing in recent years, with a greater impact on the health and economy of developing countries than developed countries [1]. According to [2] access to sufficient amounts of safe and nutritious food is key to sustaining life and promoting good health and unsafe food containing harmful bacteria, viruses, parasites or chemical substances, causes more than 200 diseases ranging from diarrhea to cancers. An estimated 600 million (almost 1 in 10 people in the world) fall ill after eating contaminated food and 420 000 die every year, including 125 000 children under the age of 5 years [2]. Common foods such as bread, beans, pasta, rice with sauce and powdered milk are common dishes suited as well as to many low-income people as to living conditions in large cities. These foods refer to street foods that play an important role in developing countries such as Burkina Faso. According to the [3], Street foods are ready-to-eat foods and beverages prepared and/or sold by vendors and hawkers especially in streets and other similar public places. This definition emphasizes the retail location on the street, with foods sold from pushcarts, bicycles, baskets or balance poles, or from stalls that do not have four permanent walls [4]. These street foods feed millions of people daily with a wide variety of ready-to-eat foods and beverages sold and sometimes prepared in the street or public places, relatively cheap and easily accessible [5, 6]. Although in developing countries the informal food vending sector has in recent years grown into a lucrative trade that competes with the formal sector, ignorance in regard to and inadequate knowledge of food handling practices, together with a lack of formal education, have prevailed among the majority of informal food handlers [7, 8]. Food can serve as ideal culture medium for the growth of microorganisms which can cause decomposition, spoilage and also serve as a vehicle for transmission of food borne illnesses [9]. The food production sites are generally located either in living spaces or near workplaces or on sale sites. The majority of these vending sites lacks basic infrastructure and services such as potable running water and waste disposal facilities, hand and dishwashing water is usually insufficient and often reused, sometimes without soap, waste water is discarded in the street and garbage often disposed of in the vicinity of the stall [7, 10].

Street food quality is a concern around the world. Studies conducted in many countries such as Mozambique [11], Malaysia [8, 12], Brazil [13], Bangladesh [14], Kenya [15] and so on have been reported. In developing countries, street foods are driven by men and women that knowledge and expertise in food handling are often limited and they often engage in street food mainly to escape poverty, especially as little start-up capital is required [3]. However, it is important to consider the health and safety impact of these food products, because foodborne infections are more and more frequent, hence the need for control strategies to ensure food safety and consumer protection. According to previous studies conducted by [9], foodborne illness is a major universal health issue in developing countries due to difficulties in safe guarding food from cross-contamination.

Burkina Faso is a land lock country neighbored by six countries, with which it shared almost the same habits of street food accessibility. Vending foods on the street is a common aspect of lifestyle in countries in which there are high unemployment, low salaries, limited work opportunities and limited social programs [16, 17]. Foods sanitary quality controls are often limited to large urban centers to the detriment of rural populations. This study is intended to investigate and shed light on the microbial safety of five common street foods sold at different points of sale in five localities of Burkina-Faso.

## Material and methods

### Sample collection and storage

One hundred and one samples divided into five groups including 15 bread samples, 12 for beans samples, 12 pasta samples, 19 rice with sauce samples and 43 milk powder samples were collected from November to December 2021 in five localities of Burkina Faso. Samples were purchased at local stores markets, street food and restaurants. Briefly, 100 g of each sample were taken aseptically and put in a sterile plastic bag, type Wagtech, (United Kingdom) certify ISO 9001, and sealed. Sample were kept less than 4˚C in a cooler containing ice box and carry in laboratory for analysis. Samples reached the laboratory were analyzed immediately or within the following 24 hours.

Table 1 shows the repartition of all the food samples submitted to this study.

Sampling sites include ordinary restaurants, local markets and shops. The concerned localities are Ouagadougou, Bobo-Dioulasso, Dakola, Cinkansé and Niangoloko. All samples were taken aseptically in sterile plastic bag, kept in an insulated cold box containing ice boxes or stored at room temperatures. Samples reached the laboratory are immediately analyzed or kept under 4˚C until used.

### Analysis parameters

Parameters applied to each group of samples were those recommended by the *Codex Alimentarius* for cooked foods namely, Total aerobic mesophilic flora, Coliforms, Thermotolerant coliforms, Yeast and mould, *Escherichia coli*, *Salmonella* spp. and Coagulase-positive staphylococci. In addition, Anaerobic Sulfito Reductive (ASR) bacteria, *Clostridium perfringens* and *Bacillus cereus* were investigated in milk only.

- *Microbial Analysis*, twenty-five (25) grams of each food sample were added into 225 mL of sterile buffered peptone water (Liofilchem diagnostic, Italy) and homogenized in a Bag Mixer (Interscience, France) for one minute at speed 6. Further tenfold serial dilutions were made with sterile buffered peptone water. Duplicate plates were made for each sample at each dilution under ISO 6887–1: 2017 standard methods. Microbial counts were expressed as Colony-Forming Units per gram (CFU/g).

### Evaluation of the food hygiene indicators

- *Total aerobic mesophilic flora*, were counted among all the food samples onto standard Plate Count Agar (PCA); (Conda Pronadisa, Spain) under **NF ISO 4833: 2013**. Plates were incubated at 30 ± 1˚C for 72 h. After incubation the number of colonies were counted on the

**Table 1. Subtotal of ready-to-eat food samples.**

| SAMPLES | OUAGADOUGOU | BOBO-DIOULASSO | DAKOLA | CINKANSÉ | NIANGOLOKO | QUANTITY | PERCENTAGE % |
|---|---|---|---|---|---|---|---|
| BEANS | 4 | 2 | 2 | 2 | 2 | 12 | **11.88** |
| MILK POWDER | 15 | 9 | 6 | 6 | 7 | 43 | **42.57** |
| BREAD | 4 | 3 | 2 | 3 | 3 | 15 | **14.85** |
| RICE WITH SAUCE | 6 | 4 | 3 | 3 | 3 | 19 | **18.81** |
| PASTA | 3 | 3 | 2 | 2 | 2 | 12 | **11.88** |
| TOTAL | **32** | **21** | **15** | **16** | **17** | **101** | **100** |

culture plate with less than 300 colonies. For culture plates with less than 15 colonies, averages were calculated to estimate the number of CFU/g.

- *Coliforms* were counted onto standard violet red bile lactose (VRBL) agar (Conda Pronadisa, Spain) and incubated at 37°C for 24 hours under **ISO 4832:2006**. Culture plates containing less than 150 colonies were considered. For culture plates with less than 15 colonies, averages were calculated to estimate the number of CFU/g.

- *Thermotolerant coliforms* are known to be an indicator of fecal contamination were counted onto standard violet red bile lactose (VRBL) agar (Conda Pronadisa, Spain) and incubated at 44.5 ± 0.5°C for 24 hours under **NF V60-2009**. Culture plates containing less than 150 colonies were considered. For culture plates with less than 15 colonies, averages were calculated to estimate the number of CFU/g.

- *Yeast and mould* were counted onto standard yeast extract glucose chloramphenicol (YGC) agar (HiMedia Laboaratories, India) and incubated at 25 ± 1°C for 5 days following **NF V08-59:2002**. The growth of moulds was checked every day in order to avoid invading colonies. Considered culture plates for bacterial counting were less than 150 colonies. For culture plates with less than 15 colonies, averages were calculated to estimate the number of CFU/g.

## Evaluation of food safety indicators

**Escherichia coli.** *E. coli* were identified through the IMViC test from thermotolerant coliforms. Briefly, suspected colonies from thermotolerant coliforms were selected and subcultured on Nutrient Agar at 37°C for 24 hours. Pure cultures grown on Nutrient Agar were used for Oxidase test and determination of IMViC pattern (indole production, methyl red reaction, Voges Proskauer and citrate utilization test) under the Standard Procedures for food Analysis. Positives colonies were transferred into Levine Eosin Methylene Blue Agar (EMB) (France), which was incubated at 37 ± 1°C for 24 hours. Colonies with green metallic sheen were considered to be *Escherichia coli*. *Escherichia coli* **ATCC 8739** was used as positive control for all analyses.

**Salmonella spp.** *Salmonella* species were investigated according to the standard—Horizontal method for detection of *Salmonella spp* **ISO 6579–1:2017**. Briefly, the non-selective enrichment was done by adding 25 g of each sample into 225 mL buffered peptone water (Liofilchem diagnostic, Italy) and homogenized in a Bag Mixer (Interscience, France). Incubation was done at 37°C for 18 to 20 hours. The selective enrichment step was performed onto both tetrathionate (Müller-Kauffman) (Liofilchem diagnostic, Italy) and Rappaport Vassiliadis Soy (Difco laboratories) broths incubated respectively at 37 ± 1°C and 42 ± 1°C for 18 to 20 hours. A brilliant green at 0.95% was added to the selective media Tetrathionate broth in order to inhibit the growth of Gram-positive bacteria. Selective isolations were performed onto Xylose Lysine Deoxycholate (HiMedia Laboratories, India) and *Salmonella-Shigella* (HiMedia Laboaratories, India) agars. Five suspected colonies of each sample were streaked onto nutrient agar and were performed using API 20E (BioMérieux, France) test for biochemical confirmation. *Salmonella typhimurium* **(ATCC 14028)** and *Salmonella enteritidis* **(ATCC 13076)** was used as positive control. The Key biochemical tests included the fermentation of glucose, negative urease reaction, lysine decarboxylase, negative indole test, $H_2S$ production, and fermentation of dulcitol [18].

**Coagulase-positive staphylococci.** Staphylococci were isolated under **ISO 6888–3:2003**. Briefly, 25 g of each food sample were dissolved in 225 ml of Peptone Water (Liofilchem diagnostic, Italy). and homogenized in a Bag Mixer (Interscience, France). A loop of 100μl of each

sample were then spread onto Baird Parker Agar (BP) supplemented with egg-yolk tellurite emulsion (HIMEDIA) and incubated under aerobic conditions at 37˚C for 24 and 48 h. The samples producing typical colonies (grey-black, surrounded by a dull halo) were considered. Biochemical confirmation to determine whenever these colonies are Coagulase positive was performed using rabbit lyophilized plasma.

**Anaerobic Sulfito Reductive (ASR) bacteria and *Clostridium perfringens*.** ASR were isolated under **ISO 15213:2003**. For this purpose, 25 g of each food sample were dissolved in 225 ml of Peptone Water (Liofilchem diagnostic, Italy) and homogenized in a Bag Mixer (Interscience, France). 1 ml from each dilution were mixed with tryptose sulfite cycloserine agar and after solidification the plates were overlayed by using the same medium. After incubations at 46˚C for 18 to 20 h under anaerobic condition, characteristic colonies were isolated for biochemical confirmation of *Clostridium perfringens* under **ISO 7937:2004**. Briefly, five black colonies were picked, and each was inoculated into 10 ml of fluid thioglycolate broth. After 18 to 20 h at 37˚C, thioglycolate tubes were used to inoculated complete lactose sulfite broth containing Durham tubes and then incubated at 37˚C for 18 to 20 h. Tubes with black butt and gas in the Durham tubes are considered as *Clostridium perfringens*. *Clostridium perfringens* **ATCC 13124** was used as a positive control for all biochemicals tests.

**Bacillus cereus.** Enumeration of *Bacillus cereus* was performed by surface plating techniques of 100µl on mannitol-egg yolk-polymyxin under **ISO 7932:2020**. This media use polymyxin B as the selective agent and permit presumptive identification by the lecithinase reaction on the egg yolk and the inability of *Bacillus cereus* to catabolize mannitol. The media were incubated at 35˚C for 24 h. The number of *Bacillus cereus* was determined after enumeration of the colonies having its characteristic appearance and submitted to biochemical characterization. *Bacillus cereus* ATCC 11778 was used as a positive control.

The appreciation criteria of microbiological food quality, namely "satisfactory", "acceptable" and "not satisfactory", of all food samples are enumerated in Table 2

## Results

The results showed that out of 101 samples submitted to this study, 73.27% were satisfactory according to the criteria used, while 14.85% were acceptable and 11.88% were not satisfactory (Fig 1). Table 3 summarizes the raw results of the evaluation of the bacterial load and the presence of pathogens in all the samples submitted to this study. Only 6.93% (7 samples over 101) of the analyzed samples did not show any microorganism. The highest number of coliforms, total aerobic mesophilic flora, thermotolerant coliforms and yeast and mould were recorded in the same sample of rice with sauce in Ouagadougou and values are respectively $2x10^4$, $2.4x10^5$, $1.7x10^4$ and $1.6x10^4$. *E. coli* were found in two (02) samples of Ouagadougou namely bean and pasta. On the other hand, *Bacillus cereus* was found in four (04) samples of milk namely two in Ouagadougou and two in Cinkansé. Neither *Salmonella* nor *Clostridium perfringens* were found in any samples.

Fig 2 summarizes the overall appreciation of all samples according to cities. Ouagadougou has the highest number of samples and the highest number of not satisfactory (7.9%). Off set Dakola, all the other localities showed at least one sample that is not satisfactory.

The Fig 3 gives an overview of the appreciation of the samples according to their nature. The milk samples represent the highest number of samples, 43 out of 101 (42.57%). All samples showed at least one case that is not satisfactory. Milk powder and rice with sauce recorded the highest cases of not satisfactory at equal value, i.e., 3.96%.

Depending on the profile of the microorganism sought in the overall samples, the total aerobic mesophilic flora was present in 93.07%, yeast and mould in 24.75%, total coliforms in

**Table 2. Quality appreciation criteria of ready to eat meals (A) and conditioned milk powder (B).**

| | Parameters | Criteria m CFU/g | M (3m) CFU/g |
|---|---|---|---|
| **A** | | | |
| **Ready meals** | *Salmonella* | Absence/25g | - - - |
| | Staphylococci | $10^2$/g | $3 \times 10^2$/g |
| | Total aerobic mesophilic flora | $3 \times 10^3$/g | $9 \times 10^3$/g |
| | Coliforms | $10^3$/g | $3 \times 10^3$/g |
| | Thermotolerants Coliforms | 10/g | $3 \times 10$/g |
| | *E. coli* | Absence | - - - |
| | Yeast and Mould | $10^4$/g | $3 \times 10^4$/g |
| **B** | | | |
| **Conditioned milk powder** | *Salmonella* | Absence/25g | - - - |
| | *Staphylococcus aureus* | 10/g | 30/g |
| | Total aerobic mesophilic flora | $5 \times 10^4$/g | $1.5 \times 10^5$/g |
| | Coliforms | 1/g | 3/g |
| | Thermotolerants Coliforms | 1/g | - - - |
| | *E. coli* | Absence | - - - |
| | Yeast and Mould | 1/g | - - - |
| | ASR | 10/g | 30/g |
| | *Clostridium perfringens* | 1/g | 3/g |
| | *Bacillus cereus* | 1/g | - - - |

Source: The bacteriological criteria retained are those stipulated by the Ministerial Decree of the French Republic of December 21, 1979 relating to ready-to-eat foods and cited by [19].

m and M are respectively lower and upper the limits of appreciation.

CFU ≤ m = Satisfactory.

m < CFU ≤ M = Acceptable.

CFU > M = Not Satisfactory.

9.90%, thermotolerant coliforms in 6.93%, *E. Coli* in 1.98% and coagulase-positive staphylococci in 0.99%. *Bacillus cereus* was present in 9.30% of milk sample only. Neither *Salmonella* and anaerobic sulfito reductive (ASR) nor *Clostridium perfringens* were detected in any samples.

## Discussion

The control of foodborne pathogens is an essential measure in preventing the appearance and spread of foodborne diseases in the population [20]. According to FAO (2009), unsafe food poses global health threats, endangering everyone; infants, young children, pregnant women [2]. Foods submitted to this study feed millions of low-income persons daily and therefore must be of very good nutritional and microbiological quality.

Despite the relatively poor sale environment hygiene noticed on the field; these foods were in general, microbiologically safe as only 11.88% were not satisfactory according to the standards used. Satisfactory results of street food were reported with breakfast and snack foods in Ghana [5], with meat and chicken stews and maizemeal porridge.in South Africa [21]. Most of the unacceptable limits derived from the total viable counts where microorganisms were present in 93,07% of samples, independently of their pathogenicity and the presence of coliforms. We can hypothesize that the total viable numbers, might come from all sources surrounding the foods, mostly from air helped by wind. The highest total viable counts recorded was

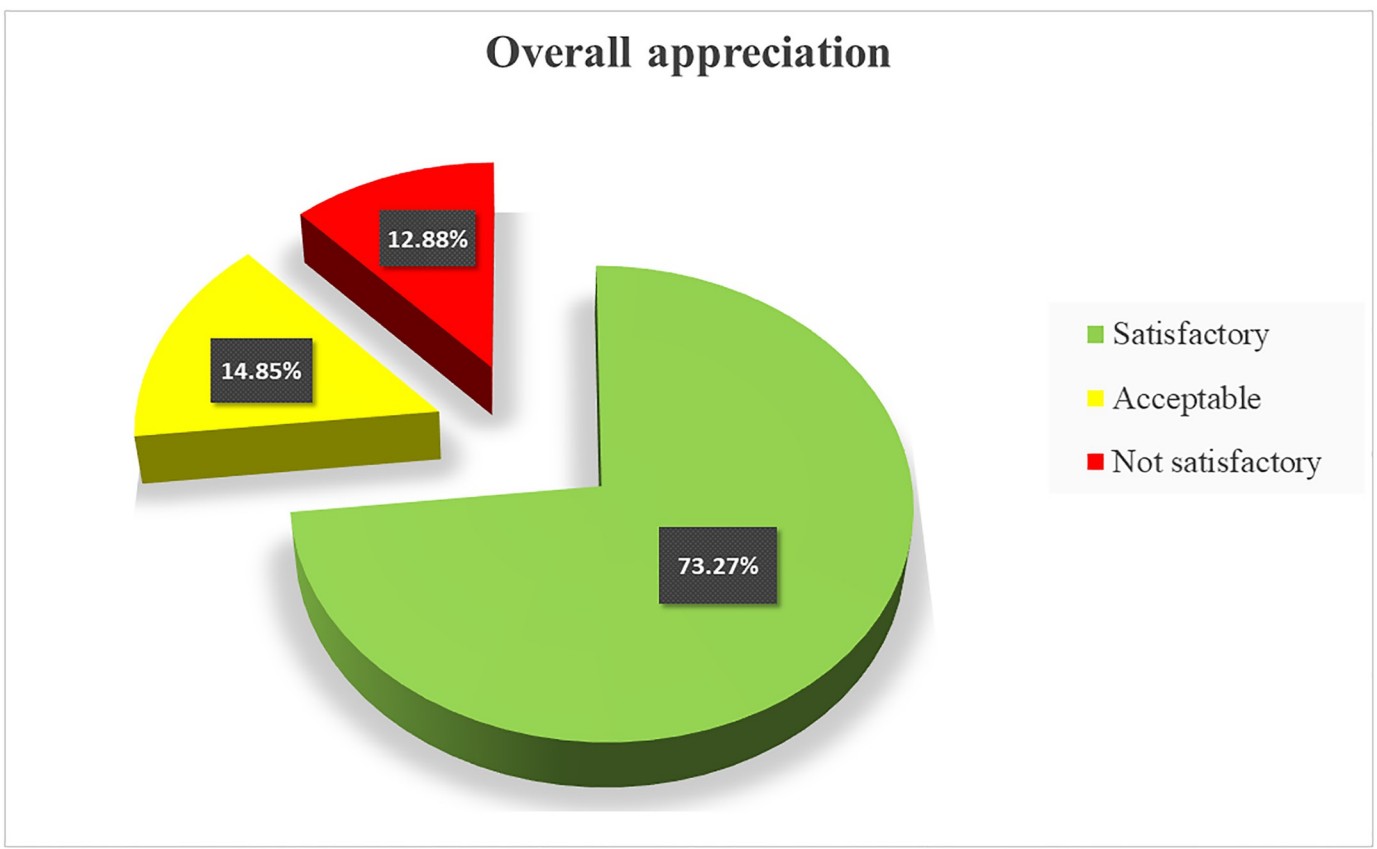

**Fig 1. The overall appreciation of samples quality.**

$2.4\times10^5$ and was found in rice with sauce. Published papers report high total viable counts in street foods from $10^5$ to $10^9$ CFU/g [13, 22, 23]. According to previous study reported by [24] many foods provide an environment conducive to microbial growth, and indicator counts in such foods may reflect the time and conditions of storage. Otherwise, total viable counts cannot be used as a safety indicator, as there is generally no correlation between its value and the presence of pathogens or their toxins [24]. Except milk samples, all food samples submitted to our study undergo heating step that should normally reduce the microorganism population. That might be one of the reasons, we found out low total viable counts as compared to other studies as well as the total flora is an indication of good conservation of hot or cold chain. Microorganisms detected seems to have post contamination origin due to food and material handling. Street foods contamination mainly occur through hands [1, 6, 11, 25]. Moreover, the presence of indicator bacteria in ready-to-eat food, although not inherently a hazard, can be indicative of poor practice that may be poor quality of raw materials or food components; undercooking; cross-contamination; poor cleaning and poor temperature and time control [26]. The presence of thermotolerants coliforms and total coliforms respectively at 6.93% and 9.90%, in the study samples confirm these finding of post-processing contamination. The thermotolerant coliforms have the same properties as the total coliforms at the difference that lactose fermentation occurred at 44.5˚C ± 0.5. However, coliforms are known to be a possible fecal contamination indicator. Their presence in food suggests the potential presence of other enteric bacteria that can be pathogens. *E. coli* were detected in 1.98% (2/101) of study samples

**Table 3. Bacterial count of sample according to localities.**

| Locality | Food type | Total Coliforms | Total aerobic mesophilic flora | Thermotolerants Coliforms | E. Coli | Coagulase-positive staphylococci | Salmonella | yeast and mould | Anaerobic Sulfito Reductive (ASR) | Clostridium perfringens | Bacillus cereus |
|---|---|---|---|---|---|---|---|---|---|---|---|
| CIN | Beans | <1E+01 | 7.0E+01 | <1E+01 | <1E+01 | <1E+02 | Absent | <1E+01 | - | - | - |
| CIN | Beans | <1E+01 | 3.0E+01 | <1E+01 | <1E+01 | <1E+02 | Absent | <1E+01 | - | - | - |
| NIA | Beans | <1E+01 | 1.3E+03 | <1E+01 | <1E+01 | <1E+02 | Absent | <1E+01 | - | - | - |
| NIA | Beans | <1E+01 | <1E+01 | <1E+01 | <1E+01 | <1E+02 | Absent | <1E+01 | - | - | - |
| DAK | Beans | <1E+01 | <1E+01 | <1E+01 | <1E+01 | <1E+02 | Absent | <1E+01 | - | - | - |
| DAK | Beans | <1E+01 | 7.8E+03 | <1E+01 | <1E+01 | <1E+02 | Absent | <1E+01 | - | - | - |
| BOB | Beans | <1E+01 | 5.0E+01 | <1E+01 | <1E+01 | <1E+02 | Absent | <1E+01 | - | - | - |
| BOB | Beans | <1E+01 | 5.0E+01 | <1E+01 | <1E+01 | <1E+02 | Absent | <1E+01 | - | - | - |
| OUA | Beans | <1E+01 | 9.0E+01 | <1E+01 | <1E+01 | <1E+02 | Absent | <1E+01 | - | - | - |
| OUA | Beans | 1.0E+02 | 1.2E+04 | 4.0E+01 | 2.0E+01 | <1E+02 | Absent | <1E+01 | - | - | - |
| OUA | Beans | <1E+01 | 5.6E+03 | <1E+01 | <1E+01 | <1E+02 | Absent | <1E+01 | - | - | - |
| OUA | Beans | <1E+01 | 1.5E+04 | <1E+01 | <1E+01 | 2.9E+02 | Absent | 4.0E+01 | - | - | - |
| CIN | Bread | 4.0E+01 | 2.5E+02 | <1E+01 | <1E+01 | <1E+02 | Absent | 1.0E+01 | - | - | - |
| CIN | Bread | <1E+01 | <1E+01 | <1E+01 | <1E+01 | <1E+02 | Absent | <1E+01 | - | - | - |
| CIN | Bread | <1E+01 | 1.0E+03 | <1E+01 | <1E+01 | <1E+02 | Absent | 2.0E+01 | - | - | - |
| NIA | Bread | <1E+01 | 2.7E+02 | <1E+01 | <1E+01 | <1E+02 | Absent | 3.0E+01 | - | - | - |
| NIA | Bread | <1E+01 | 6.5E+03 | <1E+01 | <1E+01 | <1E+02 | Absent | 3.7E+02 | - | - | - |
| NIA | Bread | <1E+01 | 8.0E+01 | <1E+01 | <1E+01 | <1E+02 | Absent | <1E+01 | - | - | - |
| DAK | Bread | <1E+01 | 2.0E+01 | <1E+01 | <1E+01 | <1E+02 | Absent | <1E+01 | - | - | - |
| DAK | Bread | <1E+01 | 8.0E+01 | <1E+01 | <1E+01 | <1E+02 | Absent | <1E+01 | - | - | - |
| BOB | Bread | <1E+01 | 7.0E+03 | <1E+01 | <1E+01 | <1E+02 | Absent | 5.1E+02 | - | - | - |
| BOB | Bread | <1E+01 | 2.7E+02 | <1E+01 | <1E+01 | <1E+02 | Absent | 1.0E+01 | - | - | - |
| BOB | Bread | <1E+01 | 7.9E+02 | <1E+01 | <1E+01 | <1E+02 | Absent | 2.0E+01 | - | - | - |
| OUA | Bread | 1.8E+02 | 1.3E+03 | 4.0E+10 | <1E+01 | <1E+02 | Absent | 2.0E+02 | - | - | - |
| OUA | Bread | <1E+01 | 6.5E+02 | <1E+01 | <1E+01 | <1E+02 | Absent | 5.3E+02 | - | - | - |

(*Continued*)

**Table 3.** (Continued)

| Locality | Food type | Total Coliforms | Total aerobic mesophilic flora | Thermotolerants Coliforms | E. Coli | Coagulase-positive staphylococci | Salmonella | yeast and mould | Anaerobic Sulfito Reductive (ASR) | Clostridium perfringens | Bacillus cereus |
|---|---|---|---|---|---|---|---|---|---|---|---|
| OUA | Bread | <1E+01 | 2.0E+01 | <1E+01 | <1E+01 | <1E+02 | Absent | <1E+01 | - | - | - |
| OUA | Bread | <1E+01 | 3.8E+02 | <1E+01 | <1E+01 | <1E+02 | Absent | 1.0E+01 | - | - | - |
| CIN | Pasta | <1E+01 | <1E+01 | <1E+01 | <1E+01 | <1E+02 | Absent | <1E+01 | - | - | - |
| CIN | Pasta | <1E+01 | 8.0E+01 | <1E+01 | <1E+01 | <1E+02 | Absent | <1E+01 | - | - | - |
| NIA | Pasta | <1E+01 | 1.1E+02 | <1E+01 | <1E+01 | <1E+02 | Absent | <1E+01 | - | - | - |
| NIA | Pasta | <1E+01 | 3.4E+02 | <1E+01 | <1E+01 | <1E+02 | Absent | <1E+01 | - | - | - |
| DAK | Pasta | 4.0E+01 | 1.4E+03 | 1.0E+10 | <1E+01 | <1E+02 | Absent | 4.2E+02 | - | - | - |
| DAK | Pasta | <1E+01 | 1.3E+02 | <1E+01 | <1E+01 | <1E+02 | Absent | <1E+01 | - | - | - |
| BOB | Pasta | <1E+01 | 1.6E+02 | <1E+01 | <1E+01 | <1E+02 | Absent | <1E+01 | - | - | - |
| BOB | Pasta | <1E+01 | <1E+01 | <1E+01 | <1E+01 | <1E+02 | Absent | <1E+01 | - | - | - |
| BOB | Pasta | <1E+01 | 2.0E+01 | <1E+01 | <1E+01 | <1E+02 | Absent | <1E+01 | - | - | - |
| OUA | Pasta | <1E+01 | <1E+01 | <1E+01 | <1E+01 | <1E+02 | Absent | <1E+01 | - | - | - |
| OUA | Pasta | <1E+01 | 1.0E+01 | <1E+01 | <1E+01 | <1E+02 | Absent | <1E+01 | - | - | - |
| OUA | Pasta | 5.0E+01 | 4.9E+04 | 3.0E+01 | 3.0E+01 | <1E+02 | Absent | <1E+01 | - | - | - |
| CIN | Rice +Sauce | <1E+01 | 6.0E+01 | <1E+01 | <1E+01 | <1E+02 | Absent | <1E+01 | - | - | - |
| CIN | Rice +Sauce | <1E+01 | 2.9 E+03 | <1E+01 | <1E+01 | <1E+02 | Absent | <1E+01 | - | - | - |
| CIN | Rice +Sauce | <1E+01 | 5.0E+01 | <1E+01 | <1E+01 | <1E+02 | Absent | <1E+01 | - | - | - |
| NIA | Rice +Sauce | <1E+01 | 2.7E+02 | <1E+01 | <1E+01 | <1E+02 | Absent | <1E+01 | - | - | - |
| NIA | Rice +Sauce | 1E+01 | 2.5E+03 | <1E+01 | <1E+01 | <1E+02 | Absent | <1E+01 | - | - | - |
| NIA | Rice +Sauce | 1.1E+02 | 2.9E+04 | <1E+01 | <1E+01 | <1E+02 | Absent | 5.1E+03 | - | - | - |
| DAK | Rice +Sauce | <1E+01 | 2.3E+02 | <1E+01 | <1E+01 | <1E+02 | Absent | <1E+01 | - | - | - |
| DAK | Rice +Sauce | <1E+01 | 2.0E+01 | <1E+01 | <1E+01 | <1E+02 | Absent | <1E+01 | - | - | - |
| DAK | Rice +Sauce | <1E+01 | 1.1E+02 | <1E+01 | <1E+01 | <1E+02 | Absent | <1E+01 | - | - | - |
| BOB | Rice +Sauce | <1E+01 | 4.0E+01 | <1E+01 | <1E+01 | <1E+02 | Absent | <1E+01 | - | - | - |
| BOB | Rice +Sauce | <1E+01 | 5.0E+01 | <1E+01 | <1E+01 | <1E+02 | Absent | <1E+01 | - | - | - |

(Continued)

**Table 3.** (Continued)

| Locality | Food type | Total Coliforms | Total aerobic mesophilic flora | Thermotolerants Coliforms | E. Coli | Coagulase-positive staphylococci | Salmonella | yeast and mould | Anaerobic Sulfito Reductive (ASR) | Clostridium perfringens | Bacillus cereus |
|---|---|---|---|---|---|---|---|---|---|---|---|
| BOB | Rice +Sauce | <1E+01 | 3.0E+04 | <1E+01 | <1E+01 | <1E+02 | Absent | 1.0E+01 | - | - | - |
| BOB | Rice +Sauce | <1E+01 | 6.7E+03 | <1E+01 | <1E+01 | <1E+02 | Absent | <1E+01 | - | - | - |
| OUA | Rice +Sauce | <1E+01 | 5.8E+03 | <1E+01 | <1E+01 | <1E+02 | Absent | <1E+01 | - | - | - |
| OUA | Rice +Sauce | <1E+01 | 5.6E+03 | <1E+01 | <1E+01 | <1E+02 | Absent | <1E+01 | - | - | - |
| OUA | Rice +Sauce | <1E+01 | 3.0E+01 | <1E+01 | <1E+01 | <1E+02 | Absent | <1E+01 | - | - | - |
| OUA | Rice +Sauce | 2.0E+04 | 2.4E+05 | 1.7E+04 | <1E+01 | <1E+02 | Absent | 1.6E+04 | - | - | - |
| OUA | Rice +Sauce | 7.9 E+02 | 5.5E+03 | 1.5E+02 | <1E+01 | <1E+02 | Absent | 1.0E+01 | - | - | - |
| OUA | Rice +Sauce | 5.0 E+01 | 2.6E+03 | 3.0E+10 | <1E+01 | <1E+02 | Absent | 2.1E+02 | - | - | - |
| NIA | Milk | <1E+01 | 1.8E+02 | <1E+01 | <1E+01 | <1E+02 | Absent | <1E+01 | <1E+01 | not detected | not detected |
| NIA | Milk | <1E+01 | 3.2E+02 | <1E+01 | <1E+01 | <1E+02 | Absent | <1E+01 | <1E+01 | not detected | not detected |
| CIN | Milk | <1E+01 | 2.3E+02 | <1E+01 | <1E+01 | <1E+02 | Absent | <1E+01 | <1E+01 | not detected | Detected |
| CIN | Milk | <1E+01 | 1.3E+02 | <1E+01 | <1E+01 | <1E+02 | Absent | <1E+01 | <1E+01 | not detected | not detected |
| CIN | Milk | <1E+01 | <1E+01 | <1E+01 | <1E+01 | <1E+02 | Absent | <1E+01 | <1E+01 | not detected | not detected |
| CIN | Milk | <1E+01 | 2.8E+02 | <1E+01 | <1E+01 | <1E+02 | Absent | <1E+01 | <1E+01 | not detected | Detected |
| CIN | Milk | <1E+01 | 4.5E+01 | <1E+01 | <1E+01 | <1E+02 | Absent | <1E+01 | <1E+01 | not detected | not detected |
| CIN | Milk | <1E+01 | 2.0E+01 | <1E+01 | <1E+01 | <1E+02 | Absent | <1E+01 | <1E+01 | not detected | not detected |
| DAK | Milk | <1E+01 | 2.0E+01 | <1E+01 | <1E+01 | <1E+02 | Absent | <1E+01 | <1E+01 | not detected | not detected |
| DAK | Milk | <1E+01 | 2.5E+02 | <1E+01 | <1E+01 | <1E+02 | Absent | <1E+01 | <1E+01 | not detected | not detected |
| DAK | Milk | <1E+01 | 2.0E+01 | <1E+01 | <1E+01 | <1E+02 | Absent | <1E+01 | <1E+01 | not detected | not detected |
| DAK | Milk | <1E+01 | 2.5E+01 | <1E+01 | <1E+01 | <1E+02 | Absent | <1E+01 | <1E+01 | not detected | not detected |
| DAK | Milk | <1E+01 | 3.0E+01 | <1E+01 | <1E+01 | <1E+02 | Absent | <1E+01 | <1E+01 | not detected | not detected |
| DAK | Milk | <1E+01 | 1.0E+01 | <1E+01 | <1E+01 | <1E+02 | Absent | <1E+01 | <1E+01 | not detected | not detected |
| OUA | Milk | <1E+01 | 4.0E+02 | <1E+01 | <1E+01 | <1E+02 | Absent | <1E+01 | <1E+01 | not detected | Detected |
| OUA | Milk | <1E+01 | 5.3E+02 | <1E+01 | <1E+01 | <1E+02 | Absent | <1E+01 | <1E+01 | not detected | not detected |
| BOB | Milk | <1E+01 | 1.5E+01 | <1E+01 | <1E+01 | <1E+02 | Absent | 1.0E+01 | <1E+01 | not detected | not detected |

(Continued)

**Table 3.** (Continued)

| Locality | Food type | Total Coliforms | Total aerobic mesophilic flora | Thermotolerants Coliforms | E. Coli | Coagulase-positive staphylococci | Salmonella | yeast and mould | Anaerobic Sulfito Reductive (ASR) | Clostridium perfringens | Bacillus cereus |
|---|---|---|---|---|---|---|---|---|---|---|---|
| BOB | Milk | <1E+01 | 6.2E+02 | <1E+01 | <1E+01 | <1E+02 | Absent | <1E+01 | <1E+01 | not detected | not detected |
| BOB | Milk | <1E+01 | 2.0E+02 | <1E+01 | <1E+01 | <1E+02 | Absent | <1E+01 | <1E+01 | not detected | not detected |
| BOB | Milk | <1E+01 | 4.0E+01 | <1E+01 | <1E+01 | <1E+02 | Absent | <1E+01 | <1E+01 | not detected | not detected |
| BOB | Milk | <1E+01 | 3.8E+02 | <1E+01 | <1E+01 | <1E+02 | Absent | <1E+01 | <1E+01 | not detected | not detected |
| BOB | Milk | <1E+01 | 1.1E+02 | <1E+01 | <1E+01 | <1E+02 | Absent | <1E+01 | <1E+01 | not detected | not detected |
| BOB | Milk | <1E+01 | 2.3E+02 | <1E+01 | <1E+01 | <1E+02 | Absent | <1E+01 | <1E+01 | not detected | not detected |
| BOB | Milk | <1E+01 | 5.5E+02 | <1E+01 | <1E+01 | <1E+02 | Absent | 1.0E+01 | <1E+01 | not detected | not detected |
| BOB | Milk | <1E+01 | 4.3E+02 | <1E+01 | <1E+01 | <1E+02 | Absent | 1.5E+01 | <1E+01 | not detected | not detected |
| OUA | Milk | <1E+01 | 3.0E+02 | <1E+01 | <1E+01 | <1E+02 | Absent | <1E+01 | <1E+01 | not detected | not detected |
| OUA | Milk | <1E+01 | 3.3E+02 | <1E+01 | <1E+01 | <1E+02 | Absent | <1E+01 | <1E+01 | not detected | not detected |
| OUA | Milk | <1E+01 | 1.1E+02 | <1E+01 | <1E+01 | <1E+02 | Absent | <1E+01 | <1E+01 | not detected | not detected |
| OUA | Milk | <1E+01 | 2.2E+02 | <1E+01 | <1E+01 | <1E+02 | Absent | <1E+01 | <1E+01 | not detected | not detected |
| OUA | Milk | <1E+01 | 4.1E+02 | <1E+01 | <1E+01 | <1E+02 | Absent | <1E+01 | <1E+01 | not detected | not detected |
| OUA | Milk | <1E+01 | 2.8E+02 | <1E+01 | <1E+01 | <1E+02 | Absent | 6.0E+01 | <1E+01 | not detected | not detected |
| OUA | Milk | <1E+01 | 3.6E+02 | <1E+01 | <1E+01 | <1E+02 | Absent | <1E+01 | <1E+01 | not detected | Detected |
| OUA | Milk | <1E+01 | 1.8E+02 | <1E+01 | <1E+01 | <1E+02 | Absent | <1E+01 | <1E+01 | not detected | not detected |
| OUA | Milk | <1E+01 | 2.0E+02 | <1E+01 | <1E+01 | <1E+02 | Absent | <1E+01 | <1E+01 | not detected | not detected |
| OUA | Milk | <1E+01 | 1.7E+02 | <1E+01 | <1E+01 | <1E+02 | Absent | <1E+01 | <1E+01 | not detected | not detected |
| OUA | Milk | <1E+01 | 1.7E+02 | <1E+01 | <1E+01 | <1E+02 | Absent | 1.0E+01 | <1E+01 | not detected | not detected |
| OUA | Milk | <1E+01 | 3.2E+02 | <1E+01 | <1E+01 | <1E+02 | Absent | <1E+01 | <1E+01 | not detected | not detected |
| OUA | Milk | <1E+01 | 2.2E+02 | <1E+01 | <1E+01 | <1E+02 | Absent | <1E+01 | <1E+01 | not detected | not detected |
| NIA | Milk | <1E+01 | 2.2E+02 | <1E+01 | <1E+01 | <1E+02 | Absent | 1.0E+01 | <1E+01 | not detected | not detected |
| NIA | Milk | <1E+01 | 1.4 E+02 | <1E+01 | <1E+01 | <1E+02 | Absent | <1E+01 | <1E+01 | not detected | not detected |
| NIA | Milk | <1E+01 | 2.2E+02 | <1E+01 | <1E+01 | <1E+02 | Absent | 1.0E+01 | <1E+01 | not detected | not detected |
| NIA | Milk | <1E+01 | 1.9E+02 | <1E+01 | <1E+01 | <1E+02 | Absent | <1E+01 | <1E+01 | not detected | not detected |

(Continued)

**Table 3.** (Continued)

| Locality | Food type | Total Coliforms | Total aerobic mesophilic flora | Thermotolerants Coliforms | *E. Coli* | Coagulase-positive staphylococci | *Salmonella* | yeast and mould | Anaerobic Sulfito Reductive (ASR) | *Clostridium perfringens* | *Bacillus cereus* |
|---|---|---|---|---|---|---|---|---|---|---|---|
| **NIA** | Milk | <1E+01 | 1.6 E+02 | <1E+01 | <1E+01 | <1E+02 | Absent | 2.0E+01 | <1E+01 | not detected | not detected |

NIA: Niangoloko; CIN: Cinkansé; DAK: Dakola; BOB: Bobo and OUA: Ouagadougou.

specifically in bean and paste samples from Ouagadougou city. Their presence in heat treated foods might therefore signify inadequate cooking or post-processing contamination. Samples collected during the holding period after the food had already been exposed to high temperature processing, any presence of *E. coli* could only be attributed to faecal contamination from the hands of food handlers and/or from contaminated working surfaces and utensils [7]. Therefore, special attention should be paid to the street food chain with reference to the 5Ms (Main œuvre, Matière, Milieu, Matériels et Méthode) as decreed in the *Codex Alimentarius*. In addition, all food handlers must pay particular attention to their personal hygiene and hand washing after handling raw food and after using the toilet.'

Coagulase-positive staphylococci were detected in a bean sample of Ouagadougou at the level of $2.6 \times 10^2$ CFU/g. Staphylococcus aureus is considered the third most important cause of

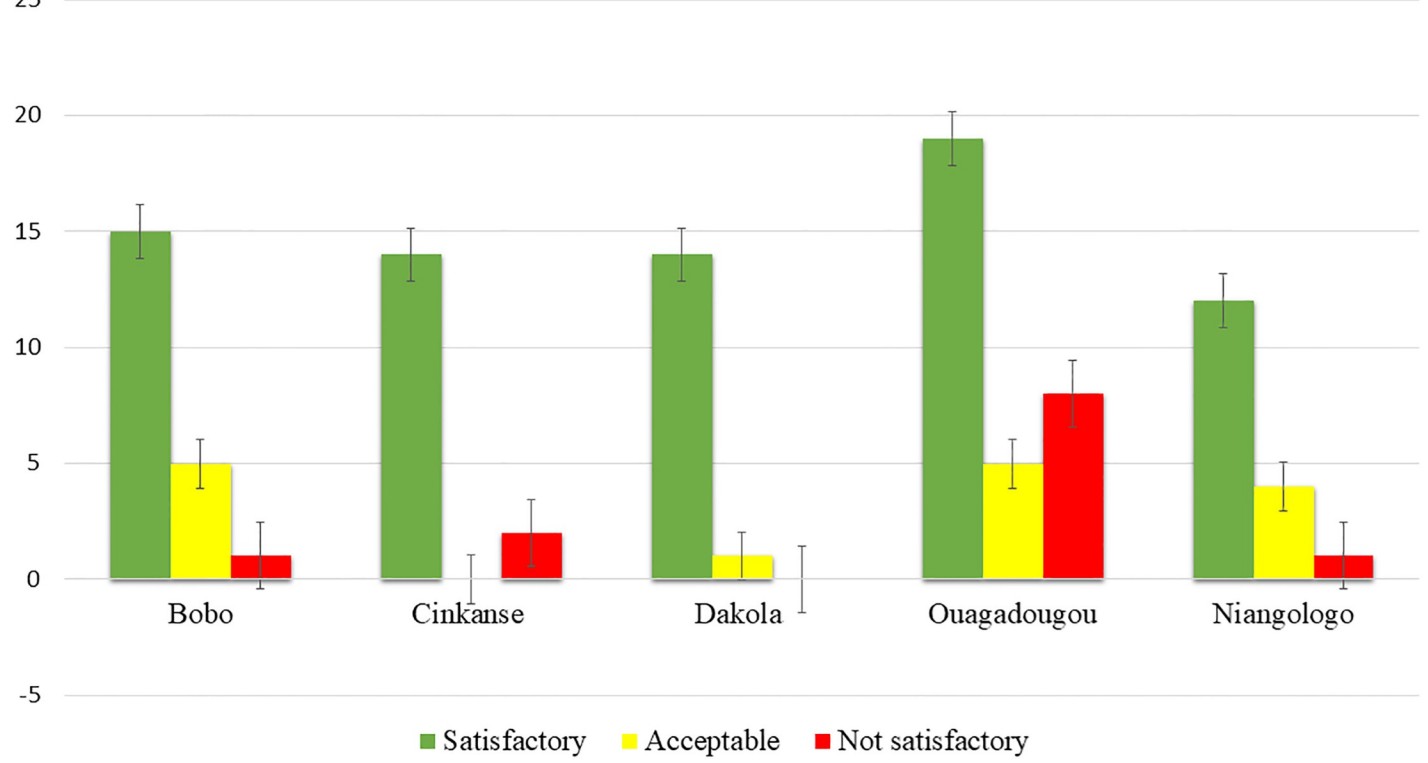

**Fig 2. Appreciation according to cities.**

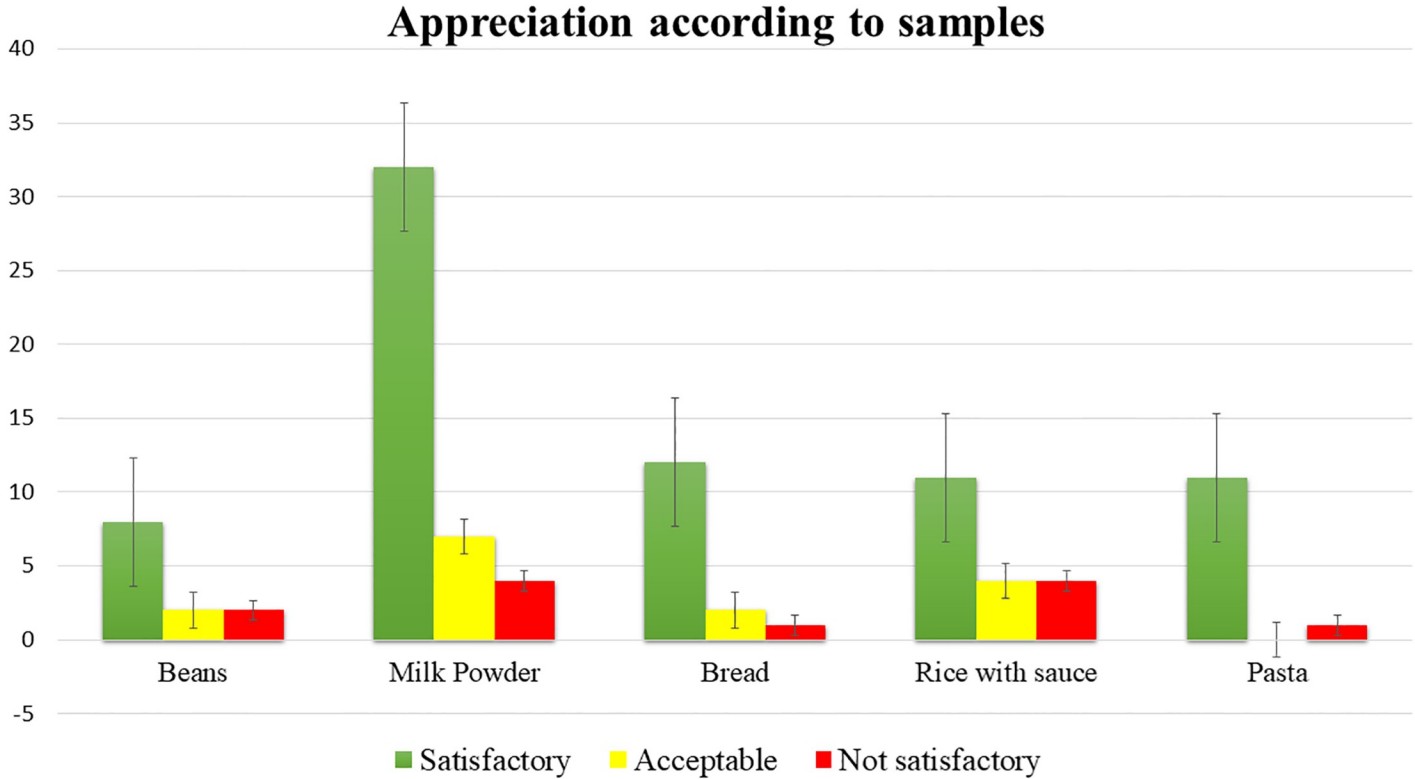

**Fig 3. Appreciation according to samples.**

food-borne diseases in the world [27]. The main reservoir of staphylococci in humans is the nostrils, although staphylococci can also be found on hand [1]. One might think that Coagulase-positive staphylococci contamination of our bean sample comes from handling, perhaps because of more frequent hand contact during preparation and serving. Staphylococcal food poisoning is one of the most common food-borne diseases in the world following the ingestion of staphylococcal enterotoxins that are produced by enterotoxigenic strains of coagulase-positive staphylococci, mainly Staphylococcus aureus [28]. Coagulase-positive staphylococci have been found in a large number of commercial foods by a wide range of investigators [7, 9, 14, 22, 23, 27] but they appear to be more present with high numeration as compared to our results.

Yeasts and molds are commonly enumerated in foods as quality indicators and they have no predictive value for the occurrence of toxigenic fungi or other pathogens [24]. They are responsible of food spoilage when these foods are exposed to ambient condition without any protection. This study exhibits 24.75% of yeast an molds contamination that load varying from $1.0\times10$ to $1.6\times10^4$ CFU/g. The presence of yeasts and molds in heat-treated foods might also has its roots in inadequate cooking, post-processing contamination, cross-contamination or even poor quality of raw materials. Previous studies find out similar results on yeast load varying from 1.2 up to 5.2 logCFU/g while assessing the microbiological quality of ethnic street foods in the Himalayas [29].

Control parameters applied to milk samples differ from those applied to the others samples as Anaerobic Sulfito-Reductive bacteria (ASR), *Clostridium perfringens* and *Bacillus cereus* were also searched. Large numbers of *Bacillus cereus* are needed to cause illness either by

releasing toxin into the food prior to consumption (emetic syndrome) or by producing a different toxin or toxins in the gut after eating the food (diarrheal syndrome) [26]. *Bacillus cereus* was isolated from only four milk powder samples (9.30%). Milk powder samples submitted to this study were somehow reconditioned by the vendors. The predominance of *Bacillus cereus* was probably due to cross contamination of bacillus spores present at the conditioning environment. Similar results were obtained in South Africa where nine meat/chicken samples (10.3%) and 6 maizemeal porridge samples (5.3%) were positive for *Bacillus cereus* [21].

Dakola, Cinkansé and Niangologo localities, which are the border post cities with high traffic and where food handlers lack of hygiene facilities (water supply, food inspectors. . .) like Ouagadougou and Bobo, were expected to have a high contamination load. Surprisingly, Ouagadougou recorded the highest rates of not satisfactory sample followed by Cinkansé. It can be hypothesized that this might be due to the high number of Ouagadougou sampling (31.68%) as compared to the other samples.

The food safety indicators such as *Salmonella*, the Anaerobic Sulfito Reductive (ASR) and *Clostridium perfringens* were not detected in any of all samples. Similar results were found in South Africa while trying to determine the health risks associated with street food vending [21]. It appears that important hygiene measures are practiced by almost all food handlers and this is very encouraging. Furthermore, the overall microbiological quality and safety of foods submitted to this research study were within the acceptable limits. Even there is no data for comparison, one might speculate that the advent of covid-19 that has profoundly destabilized developing countries people and fundamentally changed their habits and behaviors might have contributed to reduce the contamination of food through handling. Indeed, the media fuss around handwashing with soap and the use of hydro-alcoholic gels have been accepted by Burkinabe around the country. One of this positive impact might be the reduction of microbial contamination through the hands.

The presence of the different type of microorganisms and the not satisfactoriness samples observed suggested a post contamination during food handling and the possible microbial attacks propagated from the surrounding environment. It is well known that infections caused by microorganisms can be reduced by maintaining correct hand hygiene. Thus, training and education can improve the knowledge of street food handlers and play an important role in risk mitigation. Specific attention should be given for storage and packing processes specific to each food is also required, and the control of water used for utensils and hands washing. That might be one of the best ways to assure constantly a good hygienic quality of street foods.

## Conclusion

This research work highlighted that street food vendors of the study regions of Burkina Faso were able to produce relatively safe foods with low percentage of not satisfactory samples. A study that was necessary to be done, namely the quality of the meals that was normally consumes by common Burkinabe every day, especially in border areas and urban centers. The microbial quality of these foods was acceptable in general even some fecal indicators are still presents suggesting the possibility of potential presence of other enteric bacteria that can be pathogens. As the different part of the country share the same street foods habit, one could extrapolate those similar behavioral patterns may be found elsewhere within the whole country. Therefore, education and training of foods handlers is crucial to control potential food borne illness.

## Supporting information

**S1 Table. Sample GPS site.**
(DOCX)

**S1 File. Raw data.**
(XLSX)

## Acknowledgments

This work was initiated by the PAASME-UE project ("Productions et analyses des données pour améliorer la santé de la mère et de l'enfant au Burkina Faso") in collaboration with the INSP.

At the end of this work, it is important to thank:

- The National Institute of Public Health (INSP) for the scientific collaboration;

- The Health Research Ethics Committee for authorizing the conduct of this study;

- The Regional Health Directorates of the Centre, Hauts-Bassins, Cascades and Centre-East regions for authorizing the conduct of this study;

- The populations of the Centre, Hauts-Bassins, Cascades, Centre-East and Centre-South regions for their collaboration.

## Author Contributions

**Conceptualization:** Muller Kiswendsida Abdou Compaore, Stéphane Dissinviel Kpoda, Raoul Bazoin Sylvain Bazie, Marcelline Ouedraogo, Alphonse Yakoro, Souleymane Sanon, Moumouni Bande, Nicolas Barro, Elie Kabre.

**Data curation:** Muller Kiswendsida Abdou Compaore, Raoul Bazoin Sylvain Bazie, Alphonse Yakoro, Asseto Belemlougri, Naamwin-So-Bawfu Romaric Meda, Naa-Imwine Stanislas Dimitri Meda, Moumouni Bande, Nicolas Barro, Elie Kabre.

**Formal analysis:** Muller Kiswendsida Abdou Compaore, Mahamady Valian, Marie-Louise Gampene, Asseto Belemlougri, Naamwin-So-Bawfu Romaric Meda, Nicolas Barro.

**Funding acquisition:** Stéphane Dissinviel Kpoda, Souleymane Sanon, Hervé Hien, Elie Kabre.

**Investigation:** Stéphane Dissinviel Kpoda, Naamwin-So-Bawfu Romaric Meda, Moumouni Bande, Elie Kabre.

**Methodology:** Muller Kiswendsida Abdou Compaore, Raoul Bazoin Sylvain Bazie, Marcelline Ouedraogo, Alphonse Yakoro, Naamwin-So-Bawfu Romaric Meda, Naa-Imwine Stanislas Dimitri Meda, Moumouni Bande, Nicolas Barro, Elie Kabre.

**Project administration:** Fulbert Nikiema, Souleymane Sanon, Elie Kabre.

**Resources:** Souleymane Sanon, Hervé Hien, Elie Kabre.

**Software:** Muller Kiswendsida Abdou Compaore.

**Supervision:** Alphonse Yakoro, Fulbert Nikiema, Hervé Hien, Nicolas Barro.

**Validation:** Muller Kiswendsida Abdou Compaore, Stéphane Dissinviel Kpoda, Raoul Bazoin Sylvain Bazie, Marcelline Ouedraogo, Marie-Louise Gampene, Alphonse Yakoro, Asseto Belemlougri, Naamwin-So-Bawfu Romaric Meda, Nicolas Barro, Elie Kabre.

**Visualization:** Muller Kiswendsida Abdou Compaore, Stéphane Dissinviel Kpoda, Raoul Bazoin Sylvain Bazie, Fulbert Nikiema, Moumouni Bande, Nicolas Barro, Elie Kabre.

**Writing – original draft:** Muller Kiswendsida Abdou Compaore.

**Writing – review & editing:** Muller Kiswendsida Abdou Compaore, Stéphane Dissinviel Kpoda, Raoul Bazoin Sylvain Bazie, Marcelline Ouedraogo, Alphonse Yakoro, Fulbert Nikiema, Naamwin-So-Bawfu Romaric Meda, Moumouni Bande, Nicolas Barro, Elie Kabre.

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
