## [Decision Letter · Decision Letter 0]

24 Oct 2021

PONE-D-21-30741Microbiological quality assessment of five common foods sold at different points of sale in Burkina-FasoPLOS ONE

Dear Dr. COMPAORE,

Thank you for submitting your manuscript to PLOS ONE. After careful consideration, we feel that it has merit but does not fully meet PLOS ONE’s publication criteria as it currently stands. Therefore, we invite you to submit a revised version of the manuscript that addresses the points raised during the review process.

Please carefully follow the suggestion done by the reviewers

We look forward to receiving your revised manuscript.

Kind regards,

Guadalupe Virginia Nevárez-Moorillón, Ph.D.

Academic Editor

PLOS ONE

Journal Requirements:

3.  We understand that you purchased food samples from local markets for this study. In your Methods section, please provide additional details regarding the source of this material. Please provide the geographic coordinates and names of the purchase locations (e.g., stores, markets), if available, as well as any further details about the purchased items (e.g., lot number, source origin, description of appearance) to ensure reproducibility of the analyses.

"Initials of the authors who received each award: EK

Grant numbers awarded to each author: Not applicable

The full name of each funder: Europeen Union

URL of each funder website: Not applicable

"The European Union (EU) for the financing of this project"

"Initials of the authors who received each award: EK

Grant numbers awarded to each author: Not applicable

The full name of each funder: Europeen Union

URL of each funder website: Not applicable

7. Please amend either the abstract on the online submission form (via Edit Submission) or the abstract in the manuscript so that they are identical.

8. We note that Figure 1 in your submission contain map/satellite images which may be copyrighted. All PLOS content is published under the Creative Commons Attribution License (CC BY 4.0), which means that the manuscript, images, and Supporting Information files will be freely available online, and any third party is permitted to access, download, copy, distribute, and use these materials in any way, even commercially, with proper attribution. For these reasons, we cannot publish previously copyrighted maps or satellite images created using proprietary data, such as Google software (Google Maps, Street View, and Earth). For more information, see our copyright guidelines: http://journals.plos.org/plosone/s/licenses-and-copyright.

Reviewers' comments:

Reviewer's Responses to Questions

**Comments to the Author**

1. Is the manuscript technically sound, and do the data support the conclusions?

Reviewer #1: Partly

Reviewer #2: Partly

2. Has the statistical analysis been performed appropriately and rigorously? 

Reviewer #1: No

Reviewer #2: N/A

3. Have the authors made all data underlying the findings in their manuscript fully available?

Reviewer #1: No

Reviewer #2: Yes

4. Is the manuscript presented in an intelligible fashion and written in standard English?

Reviewer #1: Yes

Reviewer #2: No

5. Review Comments to the Author

Reviewer #1: The results do not support the discussion. The description of the samples should be completed, number of outlets, number of samples per outlet.....

The statistical analysis should be extended, certainly in the form of distribution (mean, median...) for a discussion on the level of hygiene of street food, by localities, by type of food...

It is also necessary to describe and justify the choice of the hygiene indicators chosen (production hygiene, hand hygiene...) to reinforce the discussion.

Comments have been attached

I am not an English speaker

Reviewer #2: Microbiological quality assessment of five common foods sold at different

points of sale in Burkina-Faso

The study is important from food safety perspectives where microbiological quality assessment has been focused on five different foods in five regions of Burkina Faso. The study is interesting though the sample size is not big enough to draw a conclusion about the food safety issue in different areas of Burkina Faso. The abstract of the manuscript is incomplete considering the findings as there is a lack of information such sampling localities, foods sampling sites (restaurants or street food), bacterial count (colony forming unit, CFU) etc. Besides, how many samples were contaminated with hygiene and safety indicator organisms? In methods, EMB agar is not selective for only E. coli, other gram-negative bacteria might grow. How did author confirm E. coli colony through EMB agar culture? There is no cfu count limit of each parameters tested in methods sections. The results were described partially and no information about the count of hygiene and safety indicators in text. The abstract shows that 11.88% foods are satisfactory in terms of microbiological standards where the results show 24% food samples were contaminated with yeast and molds, which is contradictory. In my opinion, this manuscript is not well organized. We have observed many spelling mistakes and grammatical errors throughout the manuscripts. The following comments should be addressed throughout the manuscript:

­ All percentage data should be provided with number. e.g. 43 (13%) or 13% (n=43).

­ The word ‘satisfactory/appreciation was used in multiple places of the manuscript but there is no definition of it. Is it according to international food safety criteria or guidelines?

­ The sample collection method was not described in the manuscript, only the sites of collection and sample types were mentioned.

­ In line 20, 101 is not required to write.

­ In line 23 & 24, ‘All samples were analyzed under ISO methods’ should be ‘Samples were tested according to ISO guidelines for all parameters.

­ Sampling period was not mentioned in ‘Materials and methods’ section.

­ In line 23, ‘Bacillus cereus were checked too’ should be ‘Bacillus cereus were also tested for contamination’.

­ In line 30, recorded” should be placed in lieu of “record”

­ In line 39-42, there is no reference for “According to……diarrhea to cancers” statement.

­ Line 59: “Illnesses” should be placed in lieu of “illness”.

­ Line 66-68 “In developing countries…..capital is required” is not clear to me. Rewording is required. Drove in the sentence should be driven.

­ Lin 71, not proper style of citation. Suggestion: according to previous study or something like that.

­ Line 78: Common” should be placed in lieu of “commons

­ Line 90: are should be were.

­ In line 92, ‘Figure 1 indicated the sampling localities.’ Should be ‘Figure 1. Sample collection sites in the map of Burkina Faso.’

­ Line 97, parameters name should be mentioned. And “are” should be “were”.

­ Line 100-101, homogenized in water and again homogenized in mixer bag. How many times you homogenized?

­ In line 101, add time and rotation per minute for homogenization in bag mixer.

­ In line 102, ‘with in’ should be ‘with’.

­ Line 107, what is PCA?

­ In line 108, replace ‘was’ with ‘were’.

­ In line 108, 112, 116, 121 it was not mentioned why the plates with low number of colonies were selected.

­ In line 112, ‘Only the Petri dish’ should be ‘culture plates’

­ In line 116, ‘Only the Petri dish’ should be ‘culture plates’.

­ Line 124, what is IMViC?

­ Line 128, positive clones or colonies? What is BBLTM?

­ In line 129, remove ‘agar’ after ‘(EMB)’ and ‘France’ should be in the bracket.

­ In line 130, “was used”: should be written.

­ Footnote is missing for table 2. what is m or M? what is UFC?

­ Use ‘CFU/g’ instead of in ‘UFC/g’ throughout the manuscript according to ISO guidelines.

­ What is the use of statistical analysis in Methods section? Application of significance test described in study analysis is missing in entire manuscript. Suggestion: please remove this form methods.

­ In line 134, they mentioned sesame seeds but they are not using any sesame samples. please remove sesame.

­ In line 142, ‘purified on’ should be ‘streaked onto’.

­ In line 142, remove ‘then submitted to’ and add ‘were performed using API 20E (BioMérieux, France) test’. How many colonies were undergone for API 20E biochemical tests for each parameter. Please mention in detail in the methods section.

­ In line 144, “key” and “included” should be used instead of “Key” and “including”

­ In line 145, replace ‘H2S’ with ‘H2S’.

­ In line no. 149 pepton should be replaced with peptone.

­ In line 150, replace ‘seeded’ to ‘spread’.

­ In line 157, remove ‘full stop’ after ‘(Liofilchem diagnostic, Italy)’.

­ In line 175, it was not defined what appreciation criteria is.

­ Line 188, please mention criteria of what? is it food safety criteria according to international guidelines or regulation? which food items were found to be contaminated?

­ Line 194, it is difficult to say pathogenic e. coli without performing molecular test like PCR.

­ In Table 3, data for all samples is not necessary to present in details. Data could be presented graphically. Title of Table 3 is not complete, it should be more elaborative.

­

­ In line 205, “one sample” should be written instead of “sample one”

­ Line 210, what do mean by “the milk samples represent the highest percentage i.e., 42.57%”.

­ In line no. 211 of result section “record” should be “recorded”

­ Line 217, how 11.88% food samples were satisfactory where 24.7% were contaminated with yeast and molds.

­ Reference missing in the statement “Total….wind” from line 232-234 .

­ Line 235, 105 up to109 should be 105 to109

­ Line 236, [24] report that…. This is not appropriate style of citing. Suggestion: according to previous report or something like that.

­ In line no 239 of discussion section “Foods samples” should be “food samples”.

­ In line 239, it should be written Except instead of Excepted

­ In line 247, it will be 6.93% and 9.90%. entire manuscripts had these errors.

­ Line 250, Coliforms should be coliforms.

­ Entire manuscripts show percentage of satisfactory or appreciation. Number of samples found to be contaminated should be mentioned in the parentheses along with %.

­ Line 253, signifies should be signify.

­ Line 254, According to [6] is not proper style of referencing.

­ There was no reference against the statement “That should…..other studies”; line 241-242.

­ The rationale behind the statement “That suggest the …..pathogens” (line 251-252) was not clear.

­ Unit of the count was missing in line 262 after 2.6×102

­ In line 269 it should be written, they appear to be present in higher number….

­ In line 277, it should be written “previous studies find out….”

­ In line 284, it will be 9.30%.

­ Line 284 and 287, either word or number should be written. Ex. Four or 4. No need to write both.

­ Line 290, expertise..?

­ Line 317-319: The statement “A void that was necessary…. areas and urban center” is not clear.

­ Line 324-326: too ambitious conclusion which is difficult to evaluate from this study findings. Suggestions: removed.

­ They have stated “One might believe that the advent of covid-19 that has profoundly destabilized developing countries people and fundamentally changed their habits and behaviors might have contributed to reduce the contamination of food through handling.” from line 300-302, but they don't have any data to compare with the situation before covid 19.

------------The End-----------

6. PLOS authors have the option to publish the peer review history of their article (what does this mean?). If published, this will include your full peer review and any attached files.

Reviewer #1: No

Reviewer #2: **Yes: **Dr. Mohammed Badrul Amin

---

## [Author Response · Author response to Decision Letter 0]

18 Jan 2022

Answers 

The authors would like to deeply thanks all the PLOS ONE board member for the best platform they provide for microbiological sciences promotion. Without any favor, it will be and honor for us if this manuscript is accepted for publication in PLOS ONE.

Authors also deeply thank the two reviewers for their valuable and pertinent critics that aim to improve the quality of the manuscript. We are already satisfied with the work done and will tried to bring answers for all the remarks they pointed out. Authors hope these answers will get reviewers approval. 

Thank you very much

Reviewer 1

All observations of reviewer 1 have been taken into account

Line 210, “the milk samples represent the highest percentage i.e., 42.57%” mean that portion of milk samples over all samples is 42.57%

Line 217, how 11.88% food samples were satisfactory where 24.7% were contaminated with yeast and molds: 11.88% is the overall appreciation and 24.7% is appreciation focused in only yeast and molds.

Reviewer 2

All observations of reviewer 2 have been taken into account.

L.24-26: There is no scientific reason why the percentage of contaminated food is expressed to the nearest hundredth.

L.107, L.116...The standards mentioned have been cancelled or revised. Our laboratory still used these standards because we did not buy new standard yet.

A.185: Have you verified the homogeneity within each type of food, since the points of sale are different: we did not verify the homogeneity within each type of food and I think it will not be possible to do so.

---

## [Editor Report · Decision Letter 1]

16 Feb 2022

Microbiological quality assessment of five common foods sold at different points of sale in Burkina-Faso

PONE-D-21-30741R1

Dear Dr. COMPAORE,

We’re pleased to inform you that your manuscript has been judged scientifically suitable for publication and will be formally accepted for publication once it meets all outstanding technical requirements.

Kind regards,

Guadalupe Virginia Nevárez-Moorillón, Ph.D.

Academic Editor

PLOS ONE

---

## [Editor Report · Acceptance letter]

6 Apr 2022

PONE-D-21-30741R1 

Microbiological quality assessment of five common foods sold at different points of sale in Burkina-Faso 

Dear Dr. Compaore:

I'm pleased to inform you that your manuscript has been deemed suitable for publication in PLOS ONE. Congratulations! Your manuscript is now with our production department. 

Kind regards, 

on behalf of

Dr. Guadalupe Virginia Nevárez-Moorillón 

Academic Editor

PLOS ONE